# Recent Malignant Melanoma Epidemiology in Upper Silesia, Poland. A Decade-Long Study Focusing on the Agricultural Sector

**DOI:** 10.3390/ijerph182010863

**Published:** 2021-10-15

**Authors:** Andrzej Tukiendorf, Grażyna Kamińska-Winciorek, Marcus Daniel Lancé, Katarzyna Olszak-Wąsik, Zbigniew Szczepanowski, Iwona Kulik-Parobczy, Edyta Idalia Wolny-Rokicka

**Affiliations:** 1Department of Population Health, Wrocław Medical University, ul. Bartla 5, 51-618 Wrocław, Poland; 2Department of Bone Marrow Transplantation and Onco-Hematology, National Institute of Oncology, Gliwice, ul. Armii Krajowej 15, 44-101 Gliwice, Poland; dermatolog.pl@gmail.com; 3Department of Anesthesiology, Hamad Medical Corporation, Al Rayyan Street, Doha P.O. Box 3050, Qatar; mlance@hamad.qa; 4Department of Gynecology, Obstetrics and Oncological Gynecology, School of Medicine and Division of Dentistry in Zabrze, Medical University of Silesia, ul. Batorego 15, 41-902 Bytom, Poland; katarzynaolszak@icloud.com; 5Regional Hospital in Opole, ul. Kośnego 53, 45-372 Opole, Poland; szczepan.z@poczta.fm; 6Department of Physical Education and Physiotherapy, Opole University of Technology, ul. Prószkowska 76, 45-758 Opole, Poland; i.kulik@poczta.onet.pl; 7Praxis Für Strahlentherapie und Radioonkologie, Am Stadtwall 3, 02625 Bautzen, Germany; edyta.wolny@gmail.com

**Keywords:** spatio-temporal modeling, standardized incidence ratios, growth rates, disease clusters, risk ratios, agricultural sector

## Abstract

The aim of the present study was to create spatial and spatio-temporal patterns of cutaneous malignant melanoma (MM) incidence in Upper Silesia, Poland, using the largest MM database (<4K cases) in Central Europe, focusing on the agricultural sector. The data comprised all the registered cancer cases (C43, according to the International Classification of Diseases after the 10th Revision) between the years 2004–2013 by the Regional Cancer Registries (RCRs) in Opole and Gliwice. The standardized incidence ratios (SIRs), spatio-temporal growth rates (GRs), and disease cluster relative risks (RRs) were estimated. Based on the regression coefficients, we have indicated irregularities of spatial variance in cutaneous malignant melanoma, especially in older women (≥60), and a possible age-migrating effect of agricultural population density on the risk of malignant melanoma in Upper Silesia. All the estimates were illustrated in choropleth thematic maps.

## 1. Introduction 

Historically, malignant melanoma (MM) was a rare cancer, but in the last 50 years its incidence has grown faster than almost any other cancer [1,2,3]. In some prevailing fair-skinned populations, the annual increase reaches even a few percent [4,5,6,7,8]. However, MM incidence rates vary considerably across age, gender, ethnicity, geographical location and patterns of sun exposure [9,10,11,12]. In this setting of rising statistics, MM bears a heavy health and economic burden globally [13,14].

### 1.1. Aim of the Study

The aim of the present study is to create spatial and spatio-temporal patterns of cutaneous malignant melanoma incidence in Upper Silesia, Poland, together with timeframe geographical cold- and hotspots of the disease. Additionally, in the epidemiological analysis we connect the risk for MM incidence with the densely populated rural inhabitants who participate in the agricultural sector of the region. 

### 1.2. Upper Silesia 

Upper Silesia (2018) covers the Śląskie (=12,333 km^2^ with 36 counties and 4.501 M ppl.) and Opolskie (=9412 km^2^ with 12 counties and 984 K ppl.) regions, Poland. The Śląskie region is characterized by a higher degree of urbanization and population density, whereas rural territories (with a predominance of the population employed in the agricultural sector) dominate in the Opolskie sub-region (see Figure 1). 

Based on the website data of the Polish state Institute of Meteorology and Water Management, the average long-term total solar radiation on a horizontal surface is approx. 120 W/m^2^ and has a duration of 1600 hours per year of sunshine, which places Upper Silesia in Europe at the average level of sunniness. 

## 2. Material

The data comprise of all the registered and confirmed cancer cases of malignant melanoma (C43) according to the International Statistical Classification of Diseases and Related Health Problems after the 10th Revision (ICD-10) within the years 2004–2013 collected by the Regional Cancer Registries (RCRs) in Opole and Gliwice. During the study period, 1803 cancers were confirmed in males, and 1939 in females (this is a statistical analysis of the largest dataset ever carried out in Poland). The histogram of cutaneous MM cases in age groups of patients is shown in Figure 2.

The population numbers (= denominators) were taken from Central Statistical Office.

For the purposes of further statistical analysis, the patient population was divided into younger (<60) and older (≥60) groups according to the United Nations definition. Following the registered cancers and populations, the risk for MM was lower in younger males in comparison with (<60) females with the incidence ratio (IR) = 0.84(0.78–0.92), *p* = 0.0004, whereas in older (≥60) age groups, the risk turned towards males, IR = 1.47(1.32–1.65), *p* < 0.0001; in all the patients the risk for the sexes is equal, ie. IR = 1.00(0.94–1.06), *p* = 0.9113. 

## 3. Methods

The standardized incidence ratios (SIRs) were estimated based on the conditional autoregressive (CAR) model, providing spatial patterns of the disease [15]. SIR is a ratio between the observed number of incidence in the population and the expected number of cases (if the ratio of observed/expected morbidities is greater than 1.0 = 100%, there is said to be “excess risk”, whereas when SIR<1.0, then we are dealing with “moderate risk”). In general, based on the Bayesian approach, the resulting estimators represent a weighted compromise between the overall and a local mean of the relative rate in nearby areas. 

Cancer growth rates (GRs) in each county were estimated following the spatio-temporal modeling by [16]. Spatio-temporal modeling relates to problems where disease varies over space and time. 

The spatial clusters of the disease were detected using spatial scan statistics [17]. Routinely, the term ‘disease cluster’ is used if an unusually greater or smaller than expected number of cases of the disease occurs in a group of people living in close proximity and over a limited period of time. The risk of cancer within the detected hotspots were expressed by the relative risk (RR) with the same interpretation to SIR. All the estimates were illustrated graphically in thematic maps. 

In addition, using spatial regression coefficients, we determined the effect of the agricultural population on the risk of developing malignant melanoma in the Upper Silesia region based on the CAR model [15].

The computation was performed in the WinBUGS platform (MRC Biostatistics Unit, University of Cambridge, Cambridge, UK) [18] and SatScan software (Department of Medicine, Harvard Medical School, Boston, MA, USA) [19]. 

## 4. Results

SIRs of cutaneous malignant melanoma in age groups (<60 and ≥60) in males and females in Upper Silesia, Poland (2004–2013) are shown as combined in Figure 3 in Panels A and B, respectively. 

Following the MM registry data, we observed fairly stable and similar spatial patterns of the disease in Upper Silesia in younger groups of MM patients (Figure 3, left Panels A and B), but more ‘dynamic’ SIRs in the older (≥60) population (Figure 3, right Panels A and B), especially in females. In the latter examples, northwestern and southeastern (rural) territories represent the highest cancer rates, while the remaining middle and eastern territories (industrial) represent a moderate risk of MM. 

The cancer growth rates and means of cutaneous MM (with credible 95% intervals) for the entire region are shown separately for males and females in combined Figure 4 in Panels A and B, respectively. 

In Figure 4 (left Panels A and B), the negative MM trends were mainly observed in younger males and females (<60) in Upper Silesia (2004–2013). In this age group, steep trends across the decade were observed in separate administrative units with a regional mean of GR = 2%. Similar GR means were estimated in the older (≥60) population (see right Panels A and B in Figure 4); however, the GR trends were very dynamic in contrast. 

The thematic maps in Figure 2 with a high similarity overlap those in Figure 1 (Pearson’s linear correlations for these spatial patterns *r* = 0.89, 0.89, 0.72, and 0.97, *p* < 0.0001, respectively). Since the purely spatial approach (Figure 3) is parallel with the spatio-temporal model (Figure 4), the risk for MM in the administrative units (counties) is mostly attributable with its time growth.

The spatio-temporal disease cluster of MM analysis discovered statistical significance (*p* < 0.05), but only in the older (≥60) population. Within the similar five-year timeframes, one hotspot of the disease was detected in males and two in females, respectively, (Table 1 and Panels A and B in Figure 5). 

In both examples, the disease clusters’ locations (Panels A and B in Figure 5) with their estimated risk ratios strictly correspond with the SIRs and GRs displayed earlier (right Panels in Figure 3 and Figure 4).

Finally, the effects of the agricultural sector on the risk of developing MM in age groups are reported in Table 2.

The statistical interpretation of the SIRs reported in Table 2 is as follows: agricultural population is a statistically significant (*p* < 0.05) predictor of the cutaneous MM incidence in younger males (<60). A hypothetical difference between the extremal fractions of agricultural population in administrative units (from 0% to 100%) generates a reduction of (1–0.994^100^)*100% = 45% of the risk of MM. Simultaneously, in younger females (<60), the effect is weaker and on the border of statistical significance (*p* < 0.1), i.e., the same difference gives (1–0.997^100^)*100% = 26% MM risk reduction. However, in older men (≥60) the population in the agricultural sector does not impact incidence (*p* > 0.1), but its effect increases in older women. Arithmetically, the extremal difference between the agricultural and non-agricultural population would result in a (1.004^100^ ≈) 50% increase in risk for MM; however, the effect is on the border of statistical significance (*p* < 0.1). 

## 5. Discussion

Geostatistical analyzes of malignant melanoma using so-called modern epidemiology have recently been published in scientific journals. For example, in [20] the authors for nearly two thousand Census Districts in Utah, USA, and a few decades of the observation period (from 1966 to 2017), demonstrated the effectiveness of scan statistics [19] in determining MM disease clusters, among other cancers. At this point, we proudly confirm the practicality of this geostatistical method used in the cancer epidemiology study previously published by us in [21]. With a similar population, territory, and time as presented by our study, the geostatistical results on cutaneous MM were also published by [22]. In conclusion, the authors in their analysis of CMM incidences supported the use of disease mapping for revealing geographical and socio-demographic disparities in cancer detection.

We did not expect a scientific breakthrough in our study; however, we provide exact statistical results that are brand-new in the scientific research arena of malignant melanoma incidence. Our preliminary outcomes shown in the Material section are in line with global reports. For example, with [23], which reports that age-adjusted male/female MM IR = 1.65, similarly as [24], which points out that in men, the incidence of melanoma rises rapidly after the age of 50, and nearly two thirds of melanoma deaths are male. However, the situation is surprising when taking spatio-temporal trends into account, showing higher MM incidence rates, especially in the older age group of women (see the estimates in Figure 3). These results would not have been achieved with the use of the Bayesian method.

Our second scientific observation is that the older (≥60) groups of patients show a greater spatial variance in the annual incidence growth than the younger (<60) groups (Figure 4). It is difficult for us to explain this fact clinically, and so far we are focusing only on monitoring this observation and indicating the need to explain it in the future. Recent observations also confirm the results obtained using the scan statistics, stating disease clusters of MM incidence only in the age range of older patients (see Table 1 and Figure 5). 

The last concern that we have noticed is the probability of a ‘migrating’ risk of developing cutaneous MM with the age (from younger to older persons) of the agricultural population. This may be related to the amount of time exposed to the sun’s rays, to which the population of the agricultural sector is more vulnerable. This danger is especially evident in rural women (see the results in Table 2).

At this point, some facts may be attempted to be explained clinically. Several authors, (eg. [25,26]) suggest that MM risk is reduced with decreased exposure to ovarian hormones. What is more, [25] even noticed other risk reducing factors such as: later menarche, menstrual cycle irregularity, earlier age of natural menopause and, in women whose menopause occurred naturally, shorter length of ovulatory life. Moreover, a higher melanoma risk with a higher age at first live birth and use of oral contraceptives was also reported in [27,28,29,30]. In a previous study [24], the authors suggested that the obtained data indicated the necessity for further investigation of aging males, who have lower levels of testosterone, to see if hormones exert an effect on the immune defense against cancer. On the other hand, it should be taken into account that agricultural people are a population with deeply rooted Catholicism and respect all varieties of its tradition, including a negative attitude towards contraceptives. Such social behavior may diminish the importance of additional hormonal changes induced by the use of contraception among Silesian women. What is more, the low risk of MM in the <60 population is difficult to explain by social campaigns against sunbathing, as scientific articles report rather lower sun avoidance in younger age groups, who are mainly exposed to ultraviolet radiation [31].

Certainly, it is well-known that work in the countryside is associated with the more frequent exposure to solar radiation and its impact on the incidence of MM has been already proven (see eg. [32]). What is more, in a large cohort study [33], an increase in risk of cutaneous melanoma was greater in female farm workers. Unlike men, women experienced an increased risk of skin melanoma with SIR = 1.23 (1.05–1.43), in particular, those using pesticides on crops. Similarly other authors indicate that amongst female crop farmers there were increased risks for melanoma: HR = 1.79 (1.17–2.73), and OR = 2.7 (1.2–5.8) were reported in [34] and [35], respectively. Moreover, excess risks in female farmers were concentrated among subjects with 20 or more years of farming [35] and raised risk of melanoma in outdoor workers, including farmers connected to and concerned with the duration of work >5 years with OR = 3.02 (1.88–4.86) for melanoma in situ [36] and 1.97 (1.43–2.71) for invasive melanoma [37]. The relative risk of cutaneous melanoma was 1.31 (1.14–1.52) in farmers, and 1.22 (0.99–1.50) in outdoor non-farmer occupational exposure [36].

Additionally, in the context of sex differences in cancer incidences, the employment structure in the agricultural sector can play a significant role. In the characterization of the agricultural population in the study by [38], women constituted 56% of the surveyed group of farmers, with the percentage of farmers aged up to 60 years being 55%, while farmers over 60 years of age constituted 43%. However, the elevated incidence of cutaneous melanoma may also depend on the type of pesticides used including maneb/mancozeb and parathion, and potentially benomyl, as well as lead arsenate, compared with never being users of these products [39]. 

Moreover, different levels of malignant melanoma incidences were correlated, taking into account Austrian altitude [40] and Croatian climatic factors [41]. In our study, since the most probable disease clusters in females are both located in the lowlands (upper hotspot) and highlands (lower hotspot) of the region (see Table 1 and Figure 5, Panel B), we have excluded these effects.

At the end of the discussion on the influence of sunlight on MM, it is worth mentioning the role of vitamin D in the development of this disease and its possible statistically significant (*p* < 0.05) correlation between low serum levels of vitamin D and a low percentage of serine/threonine-protein kinase B-Raf (BRAF) mutation in shield sites of malignant melanoma (ST-MM) patients, as well as between serum levels of vitamin D and higher percentage of BRAF mutation in non-shield sites of malignant melanoma (NST-MM) [41]. This discovery may explain the worse prognosis observed in patients with MM arising in chronic or intermittently sun-exposed areas due to the absence of anti-proliferative activity of vitamin D [42].

Concluding, we trust that our geostatistical analysis of malignant melanoma contributes more revealing information than just detective studies such as [20], or more methodological studies published by [22].

## 6. Conclusions

Based on the collected epidemiological material and geostatistical methodology adopted, the following conclusions can be drawn from this study:Spatial malignant melanoma standardized incidence ratios are lower in the younger (<60) age group for both men and women; a more dynamic variation of SIRs is observed in older (≥60) women than in men.The decade spatio-temporal growth rates of malignant melanoma are more dynamic in the older (≥60) population inhabiting Upper Silesia; the GRs statistically significantly correlate with the SIR models.The five-year disease clusters of malignant melanoma with an elevated relative risk in older (≥60) men and women coincide with each other in space and time; more diverse patterns were established in women.There is a ‘migration’ of malignant melanoma risk in the agricultural population from statistically significant negative in younger (<60) men towards positive in older (≥60) women. This could have connections to economic, hormonal or physiological backgrounds.The results above indicate that females are at a higher risk of malignant melanoma than males and this could only be proved with spatio-temporal data modeling.

## 7. Limitations of the Study

The main limitation of our study is its retrospective nature. Due to the extensive epidemiological material and in order not to break down the geostatistical analysis into a larger number of subtypes, our study only considered melanomas according to the ICD-10 codes (C43). Therefore, to avoid a multiplicity of statistical results we did not distinguish between the histopathological types of cancers or their location.

## Figures and Tables

**Figure 1 ijerph-18-10863-f001:**
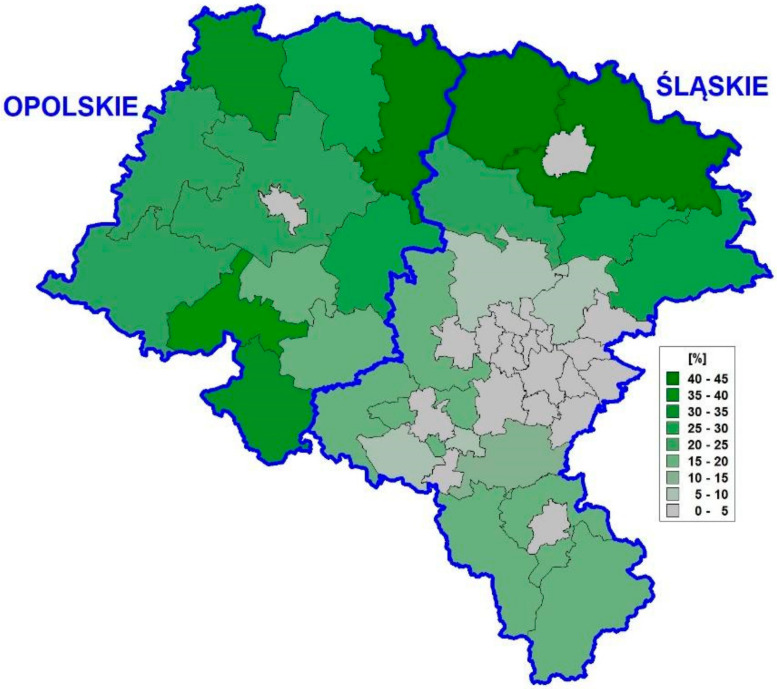
Population in agricultural sector in Upper Silesia (2004–2013).

**Figure 2 ijerph-18-10863-f002:**
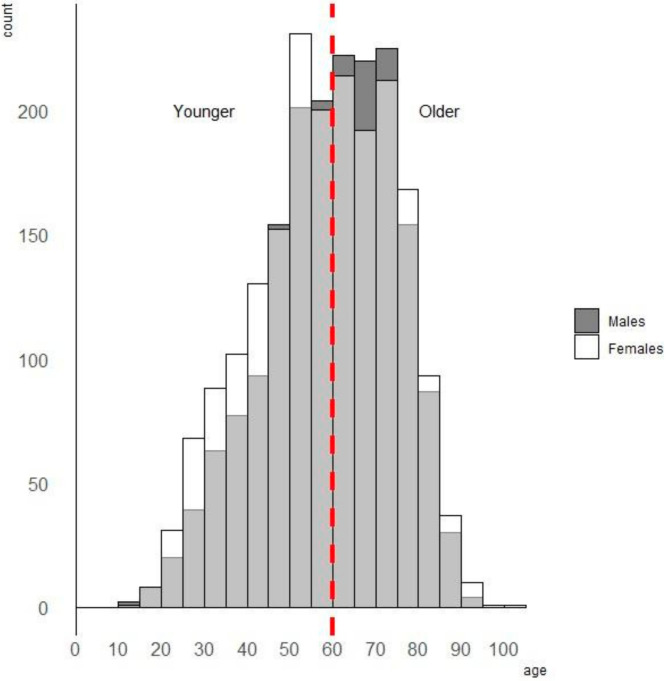
Histogram of cutaneous MM numbers in age groups of patients.

**Figure 3 ijerph-18-10863-f003:**
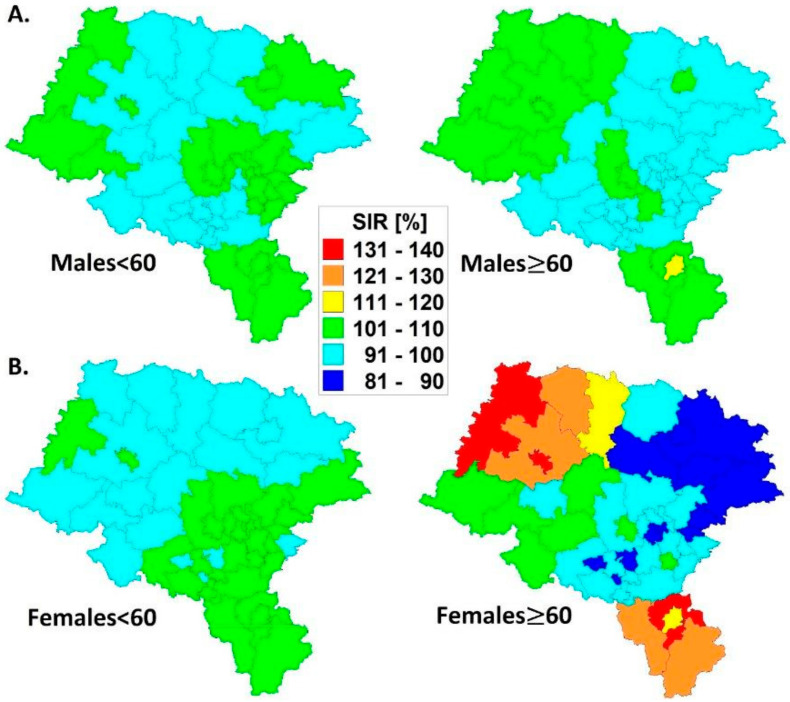
SIRs of cutaneous malignant melanoma in age groups (<60 and ≥60) in males (Panel (**A**)) and females (Panel (**B**)) in Upper Silesia, Poland (2004–2013).

**Figure 4 ijerph-18-10863-f004:**
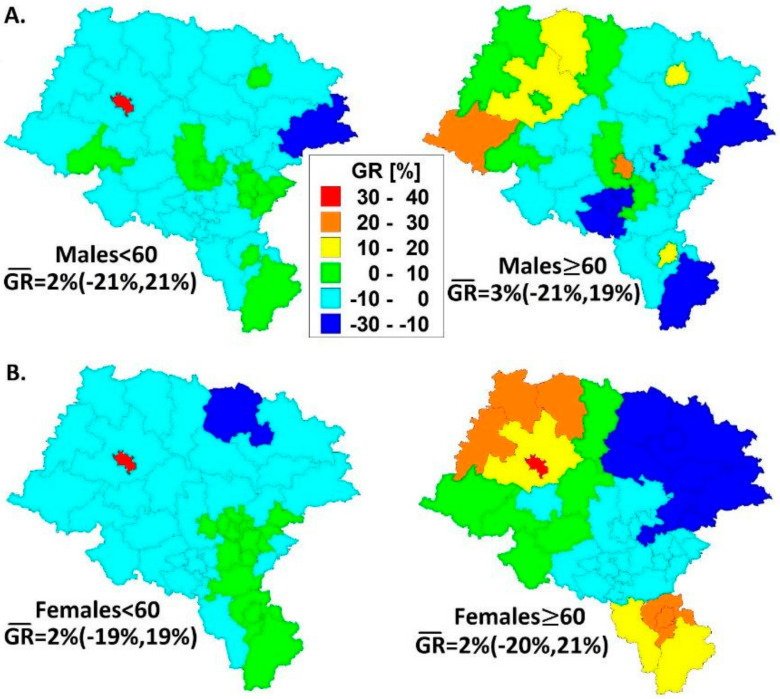
GRs of cutaneous malignant melanoma in age groups (<60 and ≥60) in males (Panel (**A**)) and females (Panel (**B**)) in Upper Silesia, Poland (2004–2013).

**Figure 5 ijerph-18-10863-f005:**
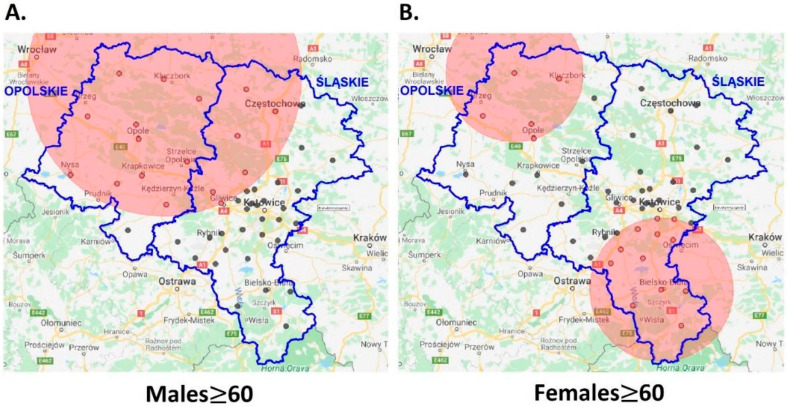
Statistically significant (*p* < 0.05) spatio-temporal cutaneous MM disease clusters (≥60) in males (Panel (**A**)) and females (Panel (**B**)) in Upper Silesia (with geographical centers of administrative units).

**Table 1 ijerph-18-10863-t001:** Statistically significant (*p* < 0.05) spatio-temporal cutaneous MM disease clusters in Upper Silesia (2004–2013) in males and females (≥60).

Population	N, E Coordinates	Radius (km)	Timeframe	RR(CI95%)	*p*-Value
Males	50.9846, 18.1490	83.82	2009–2013	1.45(1.10–1.91)	0.008
Females	49.8133, 19.0398	44.83	2008–2012	1.51(1.09–2.10)	0.014
	51.0174, 17.7593	41.90	2008–2012	1.84(1.07–3.17)	0.028

**Table 2 ijerph-18-10863-t002:** SIRs of the effect of the agricultural population on the risk for cutaneous MM in age groups (<60 and ≥60) in males and females in Upper Silesia, Poland (2004–2013).

Population	SIR	95%CI	*p*-Value
Males <60	0.994	(0.988,0.999)	0.0189
Males ≥60	0.998	(0.993,1.003)	0.2014
Females <60	0.997	(0.991,1.002)	0.0991
Females ≥60	1.004	(0.999,1.009)	0.0663

## Data Availability

All data and WinBUGS codes used in the study are available from the corresponding author upon request.

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
