# Peer review of "Recent Malignant Melanoma Epidemiology in Upper Silesia, Poland. A Decade-Long Study Focusing on the Agricultural Sector"

_ijerph, 2021, doi:10.3390/ijerph182010863_

Round 1

Reviewer 1 Report

This is an interesting and well written article about epidemiology of melanoma in Poland. In this regard, since in Poland due to latitude reasons may be interesting association with melanoma of shield sites and not shield sites also due to difference in Vitamin D, I advice to the authors to add in the discussion a speculation about this topic adding the article " Clinicopathological features, vitamin D serological levels and prognosis in cutaneous melanoma of shield-sites: an update. Med Oncol. 2015 Jan;32(1):451. doi: 10.1007/s12032-014-0451-4. Epub 2014 Dec 17. PMID: 25516505."

Please add more details regarding the incidence of acral melanoma compared with other histotypes.

Please add more informations regarding the incidence of amelanotic and hypomelanotic melanoma and extend the discussion about this topic.

Did you exclude from the analysis uveal melanoma and melanoma with unknown primary? Please explain.

Author Response

Author's Reply to the Review Report #1

Dear Reviewer, 

Thank you for your very interesting and insightful comments and we would like to emphasize that they have all been included in the revised version of our manuscript. Please also read the point-by-point responses provided by the authors to the Reviewer’s comments. At the same time, we kindly inform you that the revised manuscript with changes in red is also attached to this letter (the text was also revised linguistically by a native English specializing in editing this type of research). We remain with gratitude and respect, 

The authors. 

"This is an interesting and well written article about epidemiology of melanoma in Poland. In this regard, since in Poland due to latitude reasons may be interesting association with melanoma of shield sites and not shield sites also due to difference in Vitamin D, I advice to the authors to add in the discussion a speculation about this topic adding the article " Clinicopathological features, vitamin D serological levels and prognosis in cutaneous melanoma of shield-sites: an update. Med Oncol. 2015 Jan;32(1):451. doi: 10.1007/s12032-014-0451-4. Epub 2014 Dec 17. PMID: 25516505.""

Thanks for the suggestion to quote this interesting article. It was entered in the Discussion under item [42].

"Please add more details regarding the incidence of acral melanoma compared with other histotypes."

In our study, malignant melanomas were analyzed only according to the ICD-10 codes, therefore acral melanoma is not a histopathological term here. The term acral melanoma was changed and standardized throughout the study to be consistent with the corresponding ICD-10 number, limb melanoma. For the same reason, the study did not analyze the histopathological subtypes of melanomas, which was included in the Limitations of the study. 

"Please add more informations regarding the incidence of amelanotic and hypomelanotic melanoma and extend the discussion about this topic."

According to the above answer, the histopathological subtypes of melanomas were not analyzed, only the ICD-10 codes were used, therefore we are not able to discuss amelanotic melanomas due to the limitations of the study.

"Did you exclude from the analysis uveal melanoma and melanoma with unknown primary? Please explain."

The study excludes uveal melanoma because in our center we do not treat melanoma of the type that is mainly treated by ophthalmologists in oncological ophthalmology centers (no such cases were reported in the database). 

Reviewer 2 Report

The authors report an epidemiological analysis of melanoma trends and related risk factors in Slesia , a region belonging to the Poland state

the results presnt a limited interest as they refer to a limited area in Europe , however some interesting point come to light

in particular the authors find a difference in melanoma incidence acording to age i.e. less or more than 60 years , in particular a reduction in melanoma incidence in people <60 years

do the authors consider that this could be a consequence of more active sun protection, less sunburns and more compliance to the messages of sunscreen campaigns?

why do the authors select suc a high cut off point in terms of age, instead of a lower cut off such as 30? this could catch differences in incidence in young people ..

the authors should add a short paragraph describing the characteristis of SLesia in terms of UV exposure, UV index and temperature , moreover should define how they define rural regions

do the authors have any data about working occupations? these could represent a good marker for sun exposure

there are too many geographicla maps the authors should try to select the most representative and delete the others

Author Response

Author's Reply to the Review Report #2

Dear Reviewer, 

Thank you for your very interesting and insightful comments and we would like to emphasize that they have all been included in the revised version of our manuscript. Please also read the point-by-point responses provided by the authors to the Reviewer’s comments. At the same time, we kindly inform you that the revised manuscript with changes in red is also attached to this letter (the text was also revised linguistically by a native English specializing in editing this type of research). We remain with gratitude and respect, 

The authors. 

"The authors report an epidemiological analysis of melanoma trends and related risk factors in Slesia , a region belonging to the Poland state"

"the results present a limited interest as they refer to a limited area in Europe, however some interesting point come to light"

"in particular the authors find a difference in melanoma incidence acording to age i.e. less or more than 60 years , in particular a reduction in melanoma incidence in people <60 years"

Indeed, our study confirms that spatial malignant melanoma standardized incidence ratios are lower in the younger (<60) age group for both men and women; a more dynamic variation of SIRs is observed in older (≥60) women than in men, and this is the main conclusion of our research.

"do the authors consider that this could be a consequence of more active sun protection, less sunburns and more compliance to the messages of sunscreen campaigns?"
The reviewer drew attention to an interesting point, which we additionally discussed in the Discussion. However, it turns out that, the low risk of MM in the <60 population is difficult to explain by social campaigns against sunbathing, as scientific articles report rather lower sun avoidance in younger age groups, who are mainly exposed to ultraviolet radiation [Gavin A, Boyle R, Donnelly D, et al. Trends in skin cancer knowledge, sun protection practices and behaviours in the Northern Ireland population. Eur J Public Health. 2012;22(3):408–412] (the quoted report was included in the discussion under the number [31]).
"do the authors have any data about working occupations? these could represent a good marker for sun exposure"

Unfortunately, the analyzed database did not contain complete data on the profession, so we could not use this information in the statistical analysis. 
"why do the authors select such a high cut off point in terms of age, instead of a lower cut off such as 30? this could catch differences in incidence in young people"
For the purposes of further statistical analysis, the patient population was divided into younger (<60) and older (≥60) groups according to the United Nations definition (the information was included in the text in the Material section after the Reviewer's right comment). This age division was dictated by the beginning of retirement, common in Poland and Europe at this age. 

"the authors should add a short paragraph describing the characteristis of Silesia in terms of UV exposure, UV index and temperature , moreover should define how they define rural regions"
The information on solar radiation in Upper Silesia as well as a brief definition of the rural region was included in the Introduction after the Reviewer's right suggestions.

there are too many geographical maps the authors should try to select the most representative and delete the others
The attached geographical maps present the obtained research results complementary to the tabular ones. It is difficult for us to choose those that we should omit, because then the overall analysis of the work will be incomplete. In total, we fit into five combined drawings and we trust that we do not exceed the admissible graphic limits. Therefore, we ask for your understanding in this matter. 

Round 2

Reviewer 2 Report

the authors correctly addressed reviewer queries